# Secondary Amines from Catalytic Amination of Bio-Derived Phenolics over Pd/C and Rh/C: Effect of Operation Parameters

**Maray Ortega** [1], **Raydel Manrique** [2], **Romel Jiménez** [2], **Miriam Parreño** [3], **Marcelo E. Domine** [3] and **Luis E. Arteaga-Pérez** [1,*]

[1] Laboratory of Thermal and Catalytic Processes (LPTC), Wood Engineering Department, Faculty of Engineering, Universidad del Bio-Bio, Concepción 4030000, Chile; maray.ortega1901@alumnos.ubiobio.cl

[2] Carbon and Catalysis Laboratory (CarboCat), Department of Chemical Engineering, Faculty of Engineering, University of Concepcion, Concepción 4030000, Chile; raydelmanrique@ubiobio.cl (R.M.)

[3] Instituto de Tecnología Química (UPV-CSIC), Universitat Politècnica de València, Consejo Superior de Investigaciones Científicas, Avda. de los Naranjos s/n, 46022 Valencia, Spain

[*] Correspondence: larteaga@ubiobio.cl

**Abstract:** The production of renewable chemicals using lignocellulosic biomass has gained significant attention in green chemistry. Among biomass-derived chemicals, secondary amines have emerged as promising intermediates for synthetic applications. Here, we report a systematic study on the reductive amination of phenolics with cyclohexylamine using Pd/C and Rh/C as catalysts. The catalytic tests were performed in batch reactors under different reaction conditions (various: amine concentration (0.1–0.4 mol/L), hydrogen pressure (0–2.5 bar), temperature (80–160 °C), and substituted phenols (phenol, o-cresol, p-cresol, and methoxyphenol)) and using tert-amyl alcohol as a solvent. The experimental observations were consistent with a multi-step mechanism, where hydrogenation of phenol to cyclohexanone is followed by condensation of the ketone with cyclohexylamine to form an imine, which is finally hydrogenated to produce secondary amines. In addition, there was evidence of parallel self-condensation of the cyclohexylamine. The study also supported a limited dehydrogenation capacity of Rh/C, unlike Pd/C, which increases this capacity at higher temperatures generating a higher yield of cyclohexylaniline (up to 15%). The study of the alkylated phenols demonstrated that the nature and propensity of hydrogenation of the phenolic controls their amination. Kinetic analysis revealed reaction orders between 0.4 and 0.7 for $H_2$, indicating its dissociative adsorption. Meanwhile, phenol's order (between 1–1.8) suggests a single participation of this compound in the hydrogenation step. The order of 0.4 for cyclohexylamine suggests its participation as a surface-abundant species. The apparent activation energies derived from a power law approximation were of 37 kJ/mol and 10 kJ/mol on Pd/C and Rh/C, respectively.

**Keywords:** phenol; reductive amination; palladium; rhodium





## 1. Introduction

Secondary amines are important intermediates in the synthesis of fine chemicals and one of the most common structural components in pharmaceutical agents [1,2]. Current production routes for industrially important amines (aliphatic, aromatic, and amino alcohols) require pre-functionalized starting materials that are usually obtained from petrochemical sources [3–5]. These routes have technological drawbacks such as the generation of hazardous waste streams, low atomic and energy efficiencies [6]. Therefore, new, and more sustainable synthetic methods for producing secondary amines from lignocellulosic-derived feedstocks, are receiving attention [7]. In this sense, phenols have been reported as a naturally-available platform for amination as they can be obtained from lignin via depolymerization processes [8–13]. The reactivity of the hydroxyl group and the high dissociation energy of the C-O bonds in phenols impose the challenge of controlling the selectivity to the desired amine. Accordingly, several research groups have approached

this issue by using efficient and selective catalytic materials based on transition and noble metals (Pd, Rh, Ni, Co, Cu, etc.).

Regardless of the nature of the catalytic site, the literature presents a consensus in the proposal of reaction routes describing phenolic amination. For example, Cuypers et al. [14] studied the amination of phenol with $NH_3$ to cyclohexylamine, using hydrogen as a reducing agent, on $Ni/Al_2O_3$. They demonstrated that the reaction pathway starts with the phenol hydrogenation to cyclohexenol, which undergoes keto-enol tautomerization to cyclohexanone. Next, catalyzed condensation of cyclohexanone with ammonia occurs giving rise to cyclohexanimine. Subsequently, the imine hydrogenation leads to the formation of the desired cyclohexylamine. This route is consistent with the proposals of previous reports on the reductive amination of phenol in liquid-phase catalyzed by various transition metals [14–16]. Also, Chen et al. [16] reported a palladium-catalyzed reductive coupling of phenols with anilines to synthesize a variety of substituted cyclohexylamines. Although this work does not discuss the steric hindrance produced by the electron-withdrawing or electron-donating substituent groups in the amine, the effect of this phenomenon was evidenced in the different product selectivities.

Tomkins et al. [17], have extensively studied the amination of phenol over noble metals (Rh, Pd, and Pt), and confirmed the consecutive formation of secondary amines (dicyclohexylamine) from the hydrogenation/dehydrogenation of the imine. Results for Pd/C were remarkably good (yield of 88%) under mild reaction conditions (0.2 M of phenol, 1.4 Eq cyclohexylamine, T = 140 °C). Nevertheless, when Pt/C and Rh/C were used under identical reaction conditions, the selectivity towards cyclohexylamine (79% and 99%, respectively) was different, indicating that the metal active sites play a relevant role in the process. Additionally, it was noted that Ni/C and Ru/C were not active for this reaction. However, in a later study, it was found that $Ni/Al_2O_3$ was effective for the same reaction, demonstrating the importance of the catalyst support for this reaction [14]. Despite their relevant results, the authors do not elaborate on how the support or metal sites affected the reaction routes (condensation, hydrogenation, or dehydrogenation).

These works showed that Rh, Pt, Ni, and Pd are active for the reductive amination of phenols, leading to different product distributions, which is explained by their capacity for enhancing different reaction steps (c.a. condensation, hydrogenation, or dehydrogenation). Interestingly, this group of metals has been recognized for their greater activation of hydrogen and phenolic substrates, as compared to more oxophilic metals such as Fe, W, Re, and Mo which tend to promote deoxygenation [18]. Moreover, it has been suggested that the nature of the metal nanoparticles influences the process by controlling the hydrogenation/dehydrogenation steps [18], but a systematic analysis of other relevant operation parameters such as reagent concentrations (phenol, amine, or hydrogen) or temperature has not yet been developed. Among these metals, Pd and Rh have shown the highest activity for hydrogen activation, which has been reported as the kinetically relevant step within the catalytic cycle [17,19].

On the other hand, the amination of alkyl-phenolics on Pd/MgO was also investigated by Liu et al. [20], indicating that hysteric hindrance and substrate-catalyst interactions affect conversions and selectivity to amines. In fact, the role of steric hindrance in amination was previously observed by Gomez et al. [21] when studying the reductive amination of aldehydes and ketones. They correlated the decrease in the rate of imine intermediate formation with the increase in the size of the substituent groups in the vicinity of the mentioned functional groups [21].

In a recent study, our group has reported the effect of metal nanoparticle size and support type ($Al_2O_3$, C, and $SiO_3$) of Pd-based catalysts on the amination of phenol with cyclohexylamine [7]. In that work all the Pd-based catalysts showed remarkable activity for reductive amination, leading to dicyclohexylamine and cyclohexylaniline. In addition, $Pd^0$ showed structural sensitivity as the specific reaction rate increased with decreasing metal nanoparticle size. On the other hand, the degree of acidity and nature of acid sites (Lewis or Bronsted) in the support were relevant for the hydrogenation, where the yield of secondary

amines was strongly favored by supports with the highest fraction of Lewis-type acidic sites (activity order: $Pd/Al_2O_3 > Pd/C > Pd/SiO_2$). However, a systematic analysis of the operational parameters as well as their kinetic implications was not included in the analysis. Considering the pending questions in the field of lignin-derived phenolics amination, the present study was performed to: (i) elucidate the effect that the reaction conditions (amine concentration, hydrogen pressure, temperature, and alkyl-substituted phenols) have on product distribution from phenol amination, over different carbon-supported catalysts (Rh/C and Pd/C), (ii) quantify the kinetic implications of changing these parameters, by proposing a simple reaction rate model that fairly represents the process. These catalysts were selected inspired by their proven activity for reductive amination processes, and due to their similarities in composition, acid properties, and nanoparticle structure.

## 2. Results and Discussion

### 2.1. Surface, Textural, and Morphologic Properties of Catalysts

Table 1, summarizes the textural properties and surface acidity of Pd/C and Rh/C, while complementary characterizations of these materials can be found in previous papers from our group [22–24].

**Table 1.** Characterization of Pd/C and Rh/C catalysts.

| Catalyst | Metal Content [a] (%wt.) | Surface Area $S_{BET}$ ($m^2g^{-1}$) | Pore Volume ($cm^3g^{-1}$) | Particle Size [b] (nm) | Total Acidity [c] ($10^{-3} \mu mol_{NH3}\ m^{-2}g^{-1}$) |
|---|---|---|---|---|---|
| Pd/C | $10.7 \pm 0.5$ | $832 \pm 1.4$ | $0.58 \pm 0.2$ | $3.3 \pm 1.1$ | $8.9 \pm 0.4$ |
| Rh/C | $5.7 \pm 1.2$ | $854 \pm 0.8$ | $0.52 \pm 0.4$ | $2.5 \pm 2.1$ | $14.1 \pm 2.3$ |

[a] Measured by Atomic Absorption Spectrometry in a Perkin Elmer 3100 instrument (Perkin Elmer, Manasquan, NJ, USA). [b] Estimated from TEM images (see Figure S1). [c] Measured by $NH_3$ TPD in a 3FLEX device (Micromeritics) equipped with a TCD and combined with a mass spectrometer (Cirrus 2, MKS Instruments).

Based on these results, it appears that both catalysts have similar surface and textural properties. For instance, they have similar Brunauer–Emmett-Teller (BET) specific surface areas, total pore volumes, and Barret–Joyner–Halenda pore size distributions, with values of 2.8 nm and 3.3 nm. As a result, both catalysts can be classified as mesoporous [25]. The Weisz–Prater number ($\Phi$WP calculated for each of the reactants ($\Phi_{WPPhenol}$ = 0.0022, $\Phi_{WPCyclohexylamine}$ = 0.0138) [7]; allows withdrawing internal mass transfer limitations in the process (see detailed calculations in [26]). Additionally, the average particle size estimated from transmission electron microscopy (TEM) was in the same order (Figure S1). The similarity between the two catalysts and the use of the same support in both of them allowed us to eliminate the effects of structural sensitivity and support-related effects during the study. As a result, any changes in activity could be attributed to the nature of the metal site (Rh or Pd).

More detailed information on the local structure and surface morphology of the metal nanoparticles was inspected by high-resolution transmission electron microscopy (HRTEM) imaging (Figure 1). Figure 1a,b shows representative ex-situ HRTEM images of the Pd/C and Rh/C catalysts, respectively. The structural parameters measured on Pd/C indicate the presence of crystallized octahedral particles corresponding to Pd, with (111) facets presenting multiple plane dislocations [27,28], and an interplanar distance of 0.2 nm. Furthermore, the results obtained for Rh/C confirm the presence of abundant hexagonal and spherical particles, with an interplanar crystal spacing of 0.24 nm, which corresponds to a lattice distance (111). The results obtained in this analysis are consistent with those reported by other authors [29,30] and, are in line with previous evidence from XRD analysis for these catalysts [24].

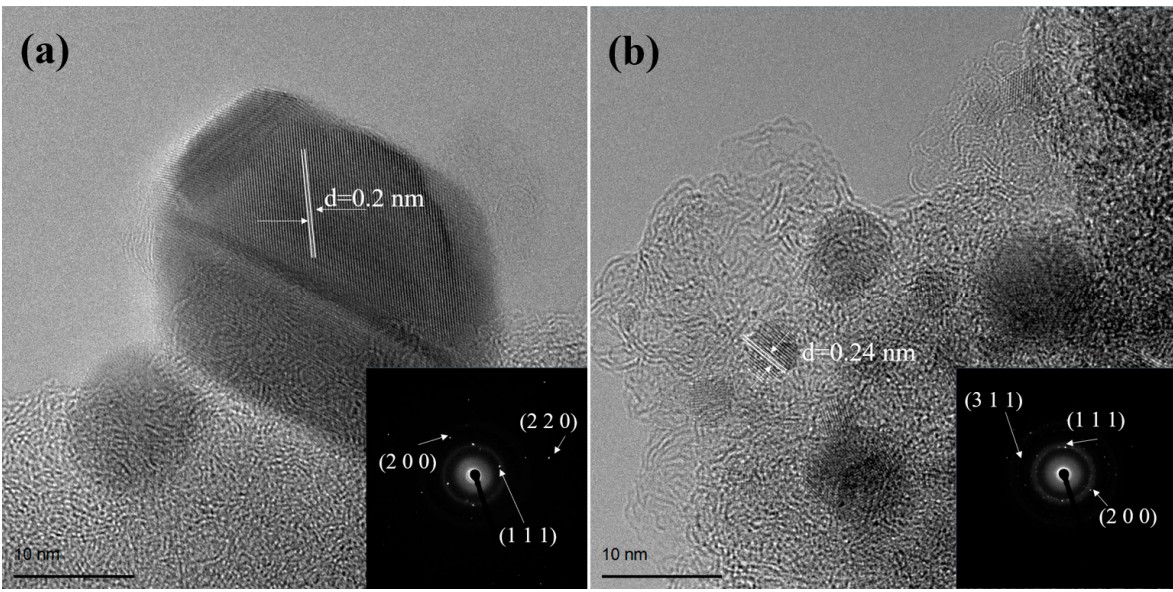

**Figure 1.** High-resolution transmission electron microscopy images of (**a**) Pd/C and (**b**) Rh/C, after reduction, respectively. Insert: electron diffraction patterns (see Figure S1 in Supporting Information for size-distribution diagrams). Arrows indicate the location of planes.

Figure 2 shows the XPS spectra to study the chemical state of the surface of commercial Pd/C and Rh/C catalysts fresh and after reduction (400 °C under $H_2$ at 45 mL/min, 2 h). The study was carried out with high-resolution spectra of the Pd $3d_{5/2}$ (Figure 2a), Rh $3d_{3/2}$ and Rh $3d_{5/2}$ (Figure 2b) regions. Deconvolution of the spectra was performed using Gaussian–Lorentzian peak shapes.

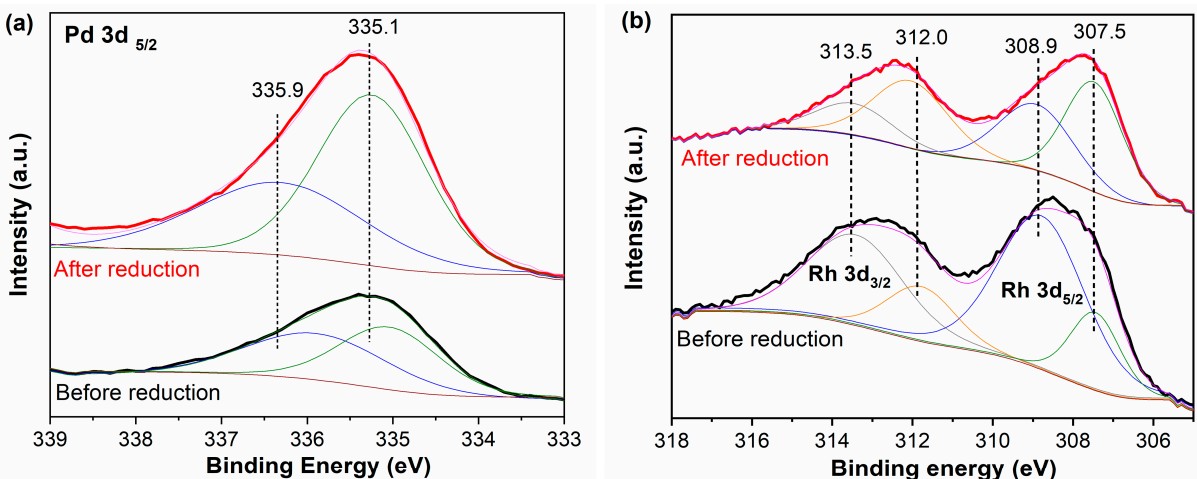

**Figure 2.** X-ray photoelectron spectroscopy spectra recorded over (**a**) Pd/C and (**b**) Rh/C catalysts before and after treatment with $H_2$. The Pd $3d_{5/2}$, Rh $3d_{3/2}$ and Rh $3d_{5/2}$ regions.

Figure 2a presents the binding energy (BE) in the Pd $3d_{5/2}$ region at 335.9 eV and 335.1 eV for the oxidized and reduced states of the metal, respectively. Meanwhile, Figure 2b shows BE values in the Rh $3d_{5/2}$ region of 308.9 and 307.5 eV, representative of the oxidized and metallic states of Rh, respectively. The same is observed in the Rh $3d_{3/2}$ region: 313.5 and 312.0 eV, representative of $Rh^{2+}$ and $Rh^0$, respectively. Considering that the BE value is not sufficient to conclude on the chemical state of the components, the Modified Auger Parameter was calculated ($\alpha'$ (eV) = binding energy (BE, eV) + kinetic energy (KE, eV)), as shown in Table S1. According to $\alpha'$, for Pd $3d_{5/2}$ indeed the BE value of 335.1 eV was representative of $Pd^0$ ($\alpha'$ = 662.4 ± 0.2 eV), while 335.9 eV best fits $Pd^{2+}$

($\alpha' = 658.2 \pm 0.5$ eV). In Rh $3d_{5/2}$ it was verified that the BE values of 307.5 and 308.9 eV correspond to $Rh^0$ ($\alpha' = 608.8 \pm 0.1$ eV) and $Rh^{2+}$ ($\alpha' = 610.3 \pm 0.1$ eV), respectively. Once the phases were identified in the spectra, the metallic and oxidized fractions of the superficial layers of the Pd and Rh nanoparticles in the Pd/C and Rh/C catalysts, respectively, were calculated. It was found that the reduction stage contributes positively to the formation of the active phase of commercial catalysts (Table S1). The surface of the nanoparticles underwent modifications, that is, in the Pd/C catalyst, $Pd^0$ went from representing 46.4% to 61.5% of the surface layers, while in Rh/C, $Rh^0$ went from 40.2% to 60.5% of the surface. However, these results confirm that the oxidized phase was not completely eliminated during the reduction protocol, representing about 40% of the most superficial layers of both Pd and Rh-supported nanoparticles. Thus, during the amination reaction, only ca. 60% of the surface of the nanoparticles represent the active phase of the catalysts, although during the reaction it is well known that these can be chemically transformed. The comparable proportion of the reduced phases suggests that the turnover frequency (TOF) computed for both catalysts will be equally impacted by the oxidation state of the metal.

## 2.2. Catalytic Experiments

### Primary Reaction Pathways

The proposal of a reaction scheme describing the amination of phenol for Pd/S (S = $Al_2O_3$, C, $SiO_2$) catalysts has been previously approached on the basis of preliminary evidence [7,19]. Here we discuss the results obtained for Rh/C as regard to that previously inspected for Pd/C but emphasizing on self-condensation of the primary amines (using cyclohexylamine and aniline). Accordingly, Table 2 summarizes the results obtained for a set of experimental conditions used to evaluate the individual roles of the catalyst, the support, the reducing agent ($H_2$), and each reactant during the amination of phenol with cyclohexylamine. Two additional experiments to assess the self-condensation of aniline were included to validate this route, and to inspect its extension as compared to the amine-ketone condensation step.

**Table 2.** Control experiments for the amination of phenol (PhOH) with cyclohexylamine (CyA) on Rh/C (30 mg). Reactions performed in 4 mL glass autoclaves at T = 120 °C, $t_R$ = 20 h, using tert-amyl alcohol as solvent.

| No. | Catalyst (Me Eqs) | PhOH (mol/L) | CyA (Eqvs.) | $H_2$ (bar) | Conversion (%) | Selectivity (%) |
|-----|-------------------|--------------|-------------|-------------|----------------|-----------------|
| 1 | n.n. | 0.20 | 1.40 | 1.5 | $X_{CyA}$ = 3% $X_{PhOH}$ = 5% | $S_{Imine}$ = 100% |
| 2 | C | 0.20 | 1.40 | 1.5 | $X_{CyA}$ = 6% $X_{PhOH}$ = 8% | $S_{CyO}$ = 20% $S_{Imine}$ = 90% |
| 3 | Rh/C | 0.0 | 1.40 | 1.5 | $X_{CyA}$ = 94% | $S_{DCyA}$ = 100% |
| 4 | Rh/C | 0.20 | 0.0 | 1.5 | $X_{PhOH}$ = 100% | $S_{CyO}$ = 100% |
| 5 | Rh/C | 0.20 | 1.40 | 0.0 | $X_{CyA}$ = 97% $X_{PhOH}$ = 24% | $S_{DCyA}$ = 78% $S_{CyPhA}$ = 2% $S_{CyO}$ = 20% |
| 6 | Rh/C | 0.20 | 1.40 | 1.5 | $X_{CyA}$ = 86% $X_{PhOH}$ = 100% | $S_{DCyA}$ = 64% $S_{CyPhA}$ = 0% $S_{Imine}$ = 36% |

The homogeneous condensation of the cyclohexylamine with phenol took place to a limited extent under noncatalytic conditions (run-1), which was confirmed by the low reaction rates ($10^{-4}$ mmol/h/$g_{cat}$) of imine formation. On the other hand, the C-support plays a role in the hydrogenation of phenol, evidenced in run-2 by the presence of cyclo-hexanone (CyO) in the products. This result indicates that phenol can be absorbed on the acidic sites of carbon to be partially hydrogenated into cyclohexanone, which is further des-

orbed by its low interaction with the support. The low conversion values obtained under this condition demonstrate that metal sites are required to enhance the hydrogenation of phenol and the further conversion of the intermediary imine to the final products. When Rh/C was introduced into the reactor containing only one of the reactants (phenol or the amine), the conversion of both cyclohexylamine (exp-3) and phenol (exp-4) was drastically increased. In the first case, the amine is self-condensated leading exclusively to the formation of dicyclohexylamine ($S_{DCyA}$ = 100%) and $NH_3$ (non-detected by GC-FID). This self-condensation of the primary amine also took place over Pd/C. However, in that case, the product distribution was different ($S_{DCyA}$ = 77% and $S_{CyPhA}$ = 23%), which suggests that $Rh^0$ and $Pd^0$ sites participate in different roles in the catalytic cycle (Figure 3a,b). In the case of Pd/C, the presence of cyclohexylaniline (CyPhA) indicates that the hydrogenation of the N=C bonds of the imine leading to dicyclohexylamine (DCyA) is followed by its dehydrogenation into CyPhA, while in Rh such step was not verified.

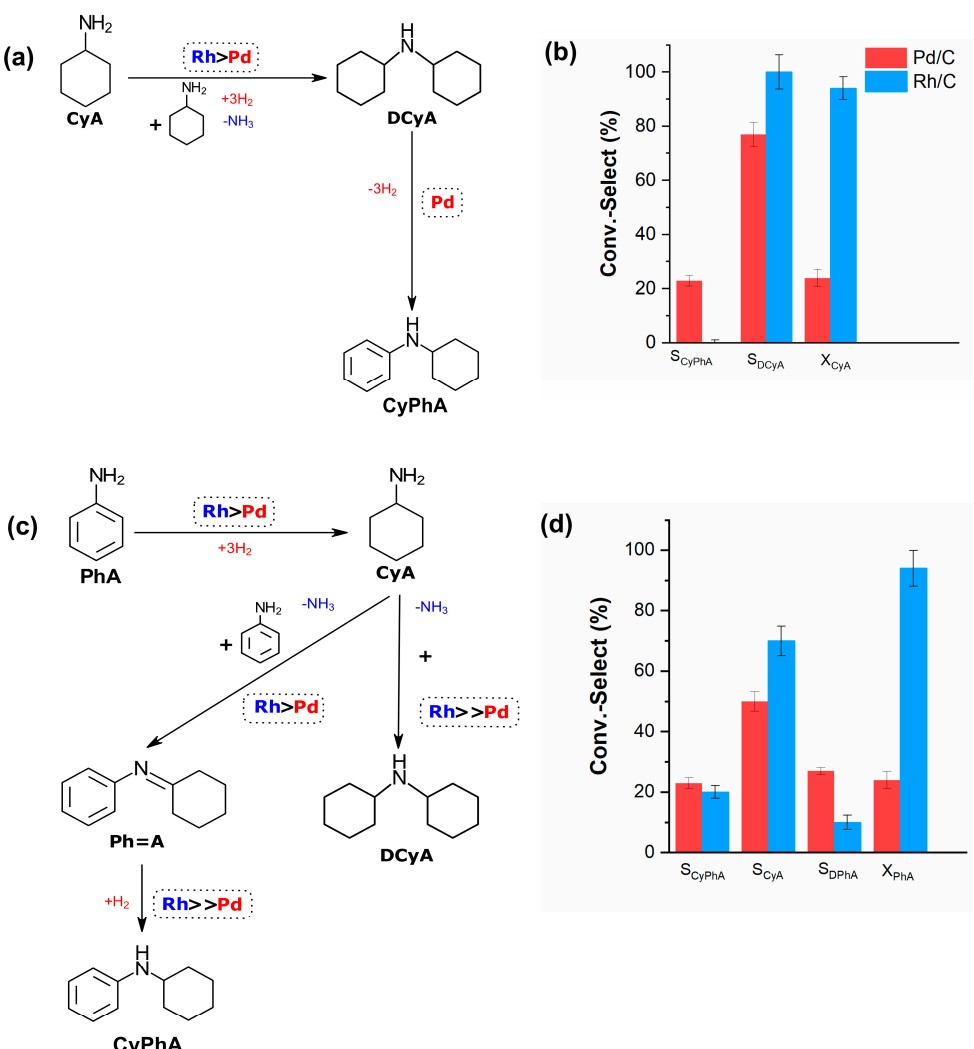

**Figure 3.** Self-condensation of primary amines (**a**) Cyclohexylamine and (**c**) Aniline on Pd/C and Rh/C with their respective concentration profiles and selectivities for (**b**) Cyclohexylamine and (**d**) Aniline. T = 120 °C, $P_{H2}$ = 1.5 bar, $C^0_{PhOH}$ = 0.2 mol/L, $C^0_{Aniline}$ = 1.4 Eqv., t = 20 h. Mcat = 0.03 g, $V_R$ = 4 mL. GC/MS data associated with these experiments is available at ESI_1 and at the mendeley data repository cited at the end of the manuscript.

The results of a replicate of this experiment but starting from aniline allows confirming the role of hydrogenation steps (Figure 3a–d).

The self-condensation of aniline shown in Figure 3 confirmed that regardless of the nature of the primary amine, the reaction is feasible over Pd/C and Rh/C. Nevertheless, major differences in the product distribution are produced by the effect of the site nature (Pd or Rh) on the hydrogenation and dehydrogenation steps. For example, in the case of Pd/C, the selectivity to dicyclohexylamine was meager (2%), while the cyclohexylaniline was 76% with the presence of traces of *N*-Cyclohexylideneaniline (Ph = A). Instead, on Rh/C, the selectivity to hydrogenated products (cyclohexylamine, 45% and dicyclohexylamine 39%) was favored to a large extent. The effect of hydrogenation sites leads to the complete conversion of phenol (exp-4) into cyclohexanone over Rh/C (Table 2). However, the complete hydrogenation into the corresponding alcohol is not completed, presumably due to the weak C-phenol interactions hindering this reaction step, as was previously witnessed by Nelson et al. [31].

Finally, exp-5 and exp-6 confirmed the need for extra hydrogen equivalents to complete the catalytic cycle over Rh/C. When the reaction was performed under a He atmosphere (exp-5), the conversion of phenol was limited to 24% with 20% selectivity to cyclohexanone. This fact, along with the high conversion of the amine (97%) and the selectivity to dicyclohexylamine, demonstrates that under an $H_2$-depleted atmosphere Rh/C promotes auto-condensation over the phenol-to-amine reaction. This is a major difference with Pd/C for which, under the same conditions, the cyclohexanone was not observed in the products while the selectivity to dehydrogenated secondary amine (CyPhA) was 61%. The exp-6 validates the effectiveness of Rh/C for the amination of phenol preferentially into dicyclohexylamine. This result is opposite to that witnessed on Pd/C, which generates a balance between DCyA and CyPhA owing to its dehydrogenation capacity [7].

In addition, tert-butylamine and hexylamine, instead of cyclohexylamine, were explored to study the extent of the N source in the reductive amination of phenols in Pd/C and Rh/C. The results of catalytic activity for the scope of the amine type are reported in the supplementary material (Table S2). Butylamine showed low yields at the reaction conditions studied (25–42%), on the other hand, heptylamine was not reactive (no secondary amines were formed), even when the reaction temperature increased (160 °C), in both catalysts. The results obtained here were similar to those reported by Junde et al. [32], who studied the amination of phenol with butylamines to N-butylcyclohexylamine in Pd/C in a continuous flow system (12 h of reaction) with acceptable yields (46%) using 20 equivalents of HCOONa as reducing agent. On the other hand, De Vos´s group investigated the nature of amines on phenol amination over 10% Pd/C and observed that when 2-heptylamine was used as an N-donating group, N-(2-heptyl)aniline (Yield = 35%) was formed as the major product, together with N-(2-heptyl)cyclohexylamine after 24 h of reaction [19]. Also, Chandan et al. [33], explored the amination of phenol with n-hexylamine on Rh-PVP catalysts and reported a moderate to low yield towards the corresponding secondary amine, even at 5 bar $H_2$ and 24 h reaction conditions. Several researchers have shown that cyclohexylamine has a greater reducing capacity for the catalytic amination of phenols [19,34]. These results allow concluding that changing the N-donating group during phenol amination has two effects, according to the specific reactivity of the amines: (i) amines can dehydrogenate less easily and (ii) the formation of secondary products can hinder product formation and decrease selectivity [19].

According to the experimental observations shown above, and inspired by previously reported literature, a reaction scheme describing the phenol with cyclohexilamine [26] over Pd/C and Rh/C is represented in Figure 4a [26].

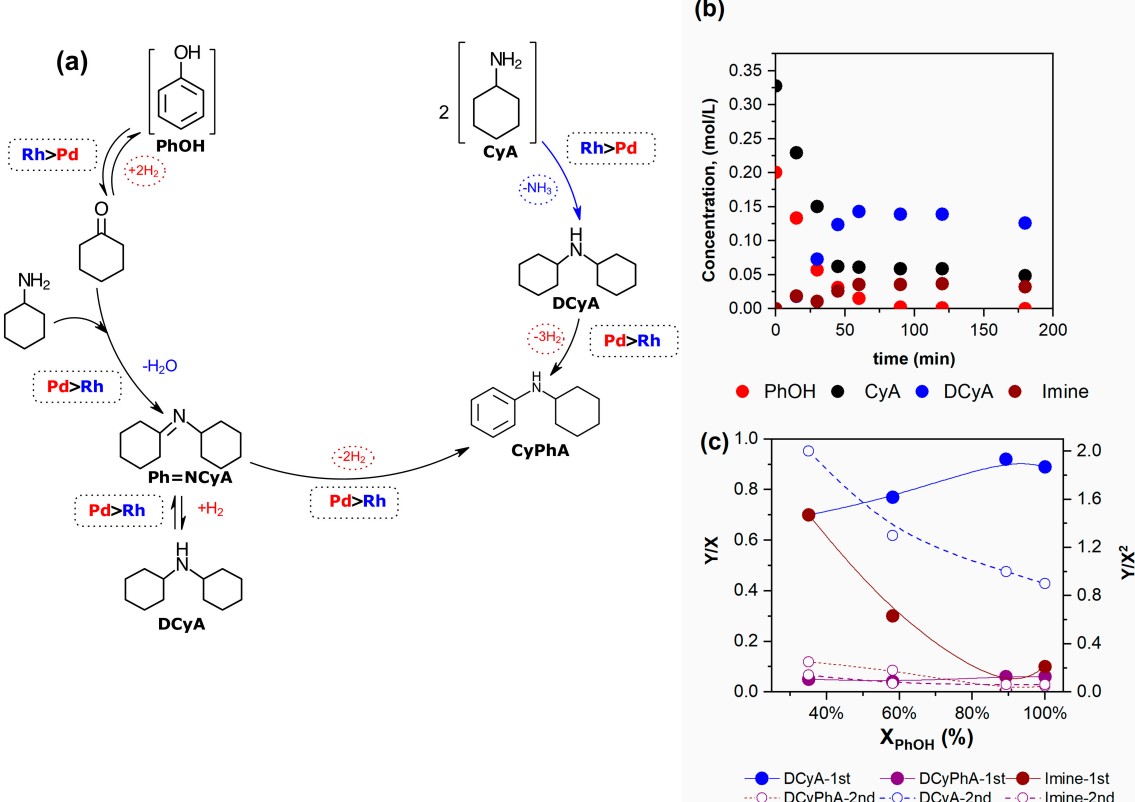

**Figure 4.** (**a**) Summarized reaction pathway for the reaction of phenol with cyclohexylamine to secondary amines over Pd/C and Rh/C. Those reaction steps enhanced by Pd or Rh are properly indicated in colors (**b**) temporal concentration profiles for phenol amination over Rh/C and (**c**) DelPlots of phenol amination over Rh/C.

The reaction is initiated by the hydrogenation of phenol into cyclohexanone, which subsequently reacts with the amine by the nucleophilic attack to form an intermediate imine. Thereafter, the imine is hydrogenated/dehydrogenated to form secondary amines, such as, dicyclohexylamine and/or cyclohexylaniline, depending on the nature of the active site. Furthermore, a self-condensation of the primary amine, parallel to the main reaction cycle could take place. This whole description confirms the complex behavior of this reaction system and indicates that the product's selectivity can be tailored by using different catalytic sites. The proposed reaction scheme was subject to a DelPlot analysis confirming the primary and secondary character of all the products (Figure 4b,c).

From the above discussion, it is clear that hydrogenation/dehydrogenation steps play important roles during phenol amination. Therefore, a systematic analysis on the effect of operational parameters on product distribution for this reaction system is provided in the upcoming section.

## 2.3. Implications of Operational Conditions and Substrate Nature

The implications of the operational parameters (amine concentration, hydrogen pressure, temperature), and the nature of phenolics on the reactivity and selectivity of Pd- and Rh-catalyzed PhOH amination are investigated.

Figure 5a,d indicate that an increase in the initial cyclohexylamine concentration from 0.15 to 0.40 mol/L has a remarkably favorable effect on the selectivity to dicyclohexylamine over Pd/C and Rh/C. In the Pd/C catalyst, the increase in the initial concentration of the amine, produces a proportional effect on phenol (20–50%) and cyclohexylamine (80–100%) conversions. In addition, the absence of cyclohexanone and the increase in the imine selectivity, for the same range suggest that the condensation step, is preferential over the self-condensation of cyclohexylamine, over Pd/C. However, for the Rh/C catalyst, the

complete conversion of phenol and decrease in the conversion of cyclohexylamine with increasing initial amine concentration, along with a decline in cyclohexanone concentration, suggests a preference for the phenol amination pathway over self-condensation. This phenomenon is further supported by the increase in imine selectivity from 10–20% under identical reaction conditions. Therefore, in both catalysts, it was found that the increase in cyclohexylamine concentration favors the phenol amination route, while at low initial concentrations of primary amine (<0.3 mol/L) the self-condensation reaction is favored. Hence, it can be suggested to work at high initial concentrations of primary amine to stimulate the phenol amination route, because the main route of the primary amine is self-condensation, that is, working at high concentrations guarantees greater availability of the amine in the system for interacting with phenol intermediates. From the quantitative point of view of active sites, the results show that despite the Pd/C catalyst containing twice as many Pd atoms on the surface than the Rh/C catalyst, its phenol conversion is remarkably lower than Rh, which can be interpreted as sites with less phenol hydrogenation capacity, but which in turn promote the amination of said intermediates to imine and later dicyclohexylamine.

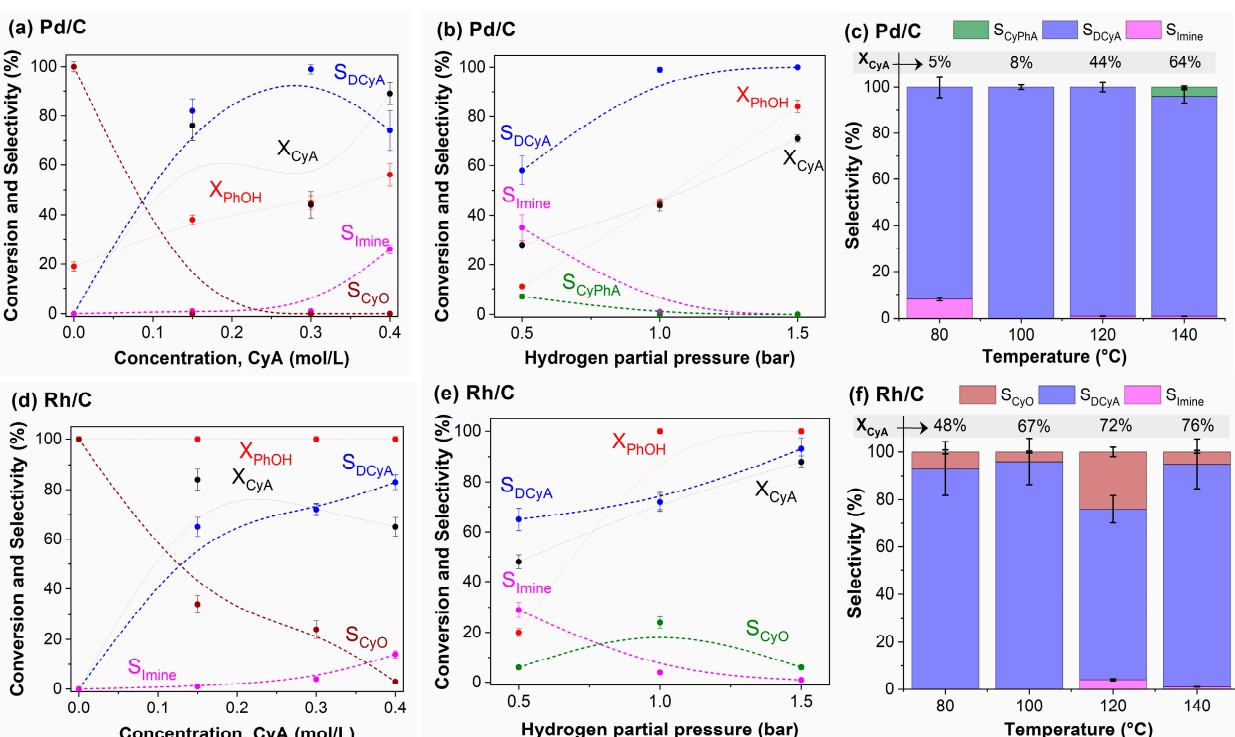

**Figure 5.** Effect of (**a**,**d**) cyclohexylamine concentration, of (**b**,**e**) H$_2$ pressure and (**c**,**f**) temperature on Pd/C and Rh/C, respectively. The reactions were carried out using H$_2$ gas (0.5–1.5 bar), He (0.5–1.5 bar), C$^0$PhOH = 0.2 mol/L, CyA (0.15–0.4 mol/L), and temperatures between 80 and 140 °C with t$_R$ = 6 h.

The preceding results suggest that there is a competition between hydrogenation reactions of the ring and C=N bonds and dehydrogenation reactions, specifically in the case of Pd, for the same active sites. Consequently, the accessibility of reducing equivalents is of vital significance to completing the catalytic cycle.

Figure 5b,e shows that an increase in H$_2$ concentration favored the conversion of the reactants towards the formation of secondary amines.

In Pd/C, a linear dependence of phenol conversion with P$_{H2}$ was observed, but this was not the case for cyclohexylamine conversion, which remained constant at P$_{H2} \geq 1$. Considering that H$_2$ is directly involved in the phenol amination pathway, it is interpreted that hydrogen promotes the formation of cyclohexanone that reacts rapidly with the primary amine to form imine, which under hydrogen limiting conditions (P$_{H2} < 1$)

is not capable to be hydrogenated into the final product. This could explain the high selectivity values to imine (ca. 35%), as well as the presence of cyclohexylaniline produced from its dehydrogenation at $0.5 < P_{H2} < 1.0$ bar. Therefore, at low $H_2$ concentrations, the self-condensation reaction of cyclohexylamine could contribute to the formation of dicyclohexylamine, while at $P_{H2} > 1$ the amination of phenols is promoted until a selectivity of 100% to dicyclohexylamine is achieved, and a phenol conversion of ca. 80%. However, in the Rh/C catalyst, cyclohexanone formation was detected regardless of the $P_{H2}$ used, which can be interpreted as the imine hydrogenation step being slower than the phenol hydrogenation on $Rh^0$, which limits the availability of active sites for the condensation step. Here, as in Pd/C, higher $H_2$ concentrations favor the phenol amination route, although in parallel it is highly probable that dicyclohexylamine continues to be formed through the primary amine self-condensation route. This Rh catalyst has a high phenol hydrogenation capacity that is not achieved in Pd, but its limited cyclohexanone amination rate represents an obstacle kinetically speaking. Meanwhile, the Pd/C catalyst promotes amination reactions and, to a lesser extent, hydrogenation/dehydrogenation reactions. Cyclohexylamine self-condensation is a reaction that will always occur regardless of the catalyst studied, but the results of reference experiments indicate that Rh/C promotes it more effectively than Pd/C ($X_{CyA}$Rh/C (94%) > $X_{CyA}$ (24%)) (Table 2 and [26]). Furthermore, this catalyst inhibits dehydrogenation reactions under $H_2$-rich conditions, while Pd/C has a more marked dehydrogenating activity, independent of $H_2$ concentration.

The reaction temperature range was studied between 80 °C and 140 °C, inspired by previous works [19,34], and the results are shown in Figure 5c,f for the Pd/C and Rh/C catalysts, respectively. At the lower temperatures 80 and 100 °C, the conversion of cyclohexylamine and phenol over Pd/C being imine and dicyclohexylamine the majority products (Figure 5c). On the other hand, for the same conditions on Rh/C, the cyclohexanone was found in the product instead the intermediary imine (Figure 5f), suggesting that hydrogenation of the imine is faster than condensation between cyclohexanone and cyclohexylamine on Rh/C. In fact, when the temperature increased up to 140 °C, the selectivity to cyclohexanone and/or imine increased at the expense of dicyclohexylamine over Rh/C and Pd/C. This indicates that the activation energy for the formation of cyclohexanone and imine is higher than that of the hydrogenation of the imine into the secondary amine. Above 140 °C, the dehydrogenation begins to be favored over Pd/C, leading to an $S_{CyPhA} = 5\%$. In the case of Rh/C, the effect of dehydrogenation was lower but still perceptible as demonstrated by the formation of cyclohexylaniline with 62% selectivity. These results demonstrate the complex behavior of the reaction mechanism taking place during the amination of phenol, where a combination of operating parameters has a significant kinetic role in product distribution. According to the reaction scheme proposed in Figure 4, the imine disproportionates into two products (cyclohexylaniline and dicyclohexylamine), where the apparent activation energy (Ea) includes the effects of the activation energies of these two parallel reactions: (i) dehydrogenation to cyclohexylaniline (Ea-1) and (ii) hydrogenation to dicyclohexylamine (Ea-2). The previous observation indicates that Ea-1 must be higher than Ea-2, in the case of Pd/C, favoring the formation of cyclohexylaniline at higher temperatures, which is also thermodynamically consistent [30]. In the upcoming section, a preliminary kinetic analysis of this reaction system is carried out.

Prior to conducting the kinetic analysis, the catalytic stability of Pd/C and Rh/C was evaluated through four and six reaction cycles, respectively (Figure S2). Pd/C maintained stability for the initial three cycles, but experienced a significant decline in activity thereafter. Similarly, Rh/C exhibited a decline in activity between the 5th and 6th cycles. The observed activity reduction could be attributed to various factors such as metal leaching and carbon deposition, or sample loss between cycles. It is important to note that pre-reduction was not conducted between cycles, and any oxidation of the catalyst surface during manipulation (washing-drying) could have led to the deactivation of the $Pd^0$ and $Rh^0$ sites. However, under the conditions used for the kinetic analysis (single cycle), these effects can be discarded.

Kinetic Implications of the Change in the Operation Parameters

Considering the kinetic effect that reactant concentrations, $H_2$ availability, and temperature have on the amination of phenol on Pd/C and Rh/C, the experimental data were interpreted by a simple kinetic model (Power Law). The modeling approach used here is the same as described in our previous papers [22,23].

Briefly, concentration vs. time profiles of phenol were correlated by polynomial regression, subject to differentiation, and extrapolated to $t \to 0$ to record the initial reaction rates.

$$r_{PhOH}^0 = \left[ \left( \frac{1}{w_{Cat} * D * \%Me} \right) \left( \frac{dC_i}{dt} \right) \right]_{t=0} \tag{1}$$

Thereafter, the effect of operation parameters on the initial rates were interpreted according to the following expression:

$$r_{PhOH}^0 = \left[ \left( \frac{1}{w_{Cat} * D * \%Me} \right) \left( \frac{dC_i}{dt} \right) \right]_{t=0} = \left( \frac{1}{w_{Cat} * D * \%Me} \right) k \left( C_{PhOH}^0 \right)^\alpha \left( C_{CyA}^0 \right)^\beta \left( C_{H2}^0 \right)^\gamma \tag{2}$$

$$k = A \cdot \exp \left[ \left( \frac{-Ea}{RT} \right) \right] \tag{3}$$

Here $C_i$ is the concentration (mol/L), wcat is the mass of the catalyst (g), D is the dispersion and %Me is the metal content of Pd or Rh in the catalyst (% wt.), Ea is the activation energy in kJ/mol, R the universal gas constant (8.31 J/K/mol) and A the pre-exponential factor, $\alpha, \beta, \gamma$ are the reaction orders for phenol, cyclohexylamine and hydrogen, respectively.

The concentration of hydrogen in the liquid phase ($C_{H2}^0$) was calculated using the Herry's law. In this case, the vapor pressure of solvent at the reaction temperature was calculated by the Antoine´s equation and the partial pressure of hydrogen was estimated by using the Dalton's law.

The reaction rate constant was determined using the Arrhenius law, which involved analyzing an Arrhenius-type plot over a temperature range of 100 °C to 140 °C. The value of k at each temperature is obtained from a co-linear regression of r° vs. kCi (Equation (2)) and then A and Ea are obtained from the adjustment of these k with the Arrhenius Eq (Equation (3)).

The reactant concentrations and hydrogen pressures varied in the same range as presented in the preceding section (i.e., 0.15 mol/L < $C^0_{CyA}$ < 0.40 mol/L, 0.15 mol/L < $C^0_{PhOH}$ < 0.40 mol/L and 1 bar < $P^0_{H2}$ < 1.5 bar). The raw data used for the analysis and the datasheets for the mathematical treatment are provided in a recently published paper [24].

The Arrhenius plots and data fitting for reaction orders are provided in the Supplementary material (Figures S3 and S4).

The resulting power-law kinetic expressions for phenol amination over Rh/C and Pd/C are presented in Equations (4) and (5), respectively.

$$r_{Rh/C}^0 = \left[ \left( \frac{1}{w_{Cat} * D * \%Me} \right) \left( \frac{dC_i}{dt} \right) \right]_{t=0} = 1 \times 10^4 \exp \left( \frac{9.7}{R \cdot T} \right) \left( C_{PhOH}^0 \right)^{1.8} \left( C_{CyA}^0 \right)^{0.41} \left( C_{H2}^0 \right)^{0.7} \tag{4}$$

$$r_{Pd/C}^0 = \left[ \left( \frac{1}{w_{Cat} * D * \%Me} \right) \left( \frac{dC_i}{dt} \right) \right]_{t=0} = 7 \times 10^6 \exp \left( \frac{37}{R \cdot T} \right) \left( C_{PhOH}^0 \right)^1 \left( C_{CyA}^0 \right)^{0.41} \left( C_{H2}^0 \right)^{0.4} \tag{5}$$

The value computed for the apparent activation energy (Ea) for phenol conversion over Rh/C the 9.7 kJ/mol, is quite lower than that reported for cyclohexanone amination over the same catalyst (55 kJ/mol) [22]. We ascribed this divergence to the hydrogenation of phenol is enhanced on $Rh^0$ sites as was demonstrated in the previous sections, thus the $E_{app}$ for the whole amination process is underestimated in this case. In the case of Pd/C the $E_{app}$ the 37 kJ/mol was slightly lower than that reported for cyclohexanone amination (48 kJ/mol) and for the N-alkylation of nitriles with alcohols (26.1–30.6 kJ/mol) [35].

The apparent reaction orders of hydrogen ranged between 0.40 and 0.70, indicating the involvement of H atoms from the dissociative adsorption of $H_2$ on a pair of metal sites. On the other hand, for phenol, the apparent order of 1–1.8 suggests its conversion via hydrogenation in a single reaction step. As for the amine, the fractional apparent order of 0.40 suggests that the adsorbed cyclohexylamine is involved in the rate-determining step (RDS) but also contributes to the site balance, as manifested by MARI on the surface.

The results indicate that the amination of phenol over Rh/C and Pd/C follows a similar reaction mechanism, where hydrogenation steps are crucial. To our knowledge, this is the first study to examine the kinetic implications of operating parameter changes in the catalytic amination of phenols. However, a more elaborated kinetic analysis is needed to determine the relative importance of each elementary reaction step in the mechanism, and to establish a mechanism-centered kinetic model.

### 2.4. Steric Effects of Alkyl-Phenolics

Different phenolics were tested to verify the effect of the nature and position of substituents on the product distribution and the general reaction descriptors (TOF and Conversion).

When Pd/C was used as the catalyst, all the phenolics were converted into their respective substituted cyclohexylamines as evidenced in Figure 6a–f. These figures were elaborated from GC/MS measurements. Detailed chromatographic information on these analyses can be found in ESI_1 and Ortega et al. [24].

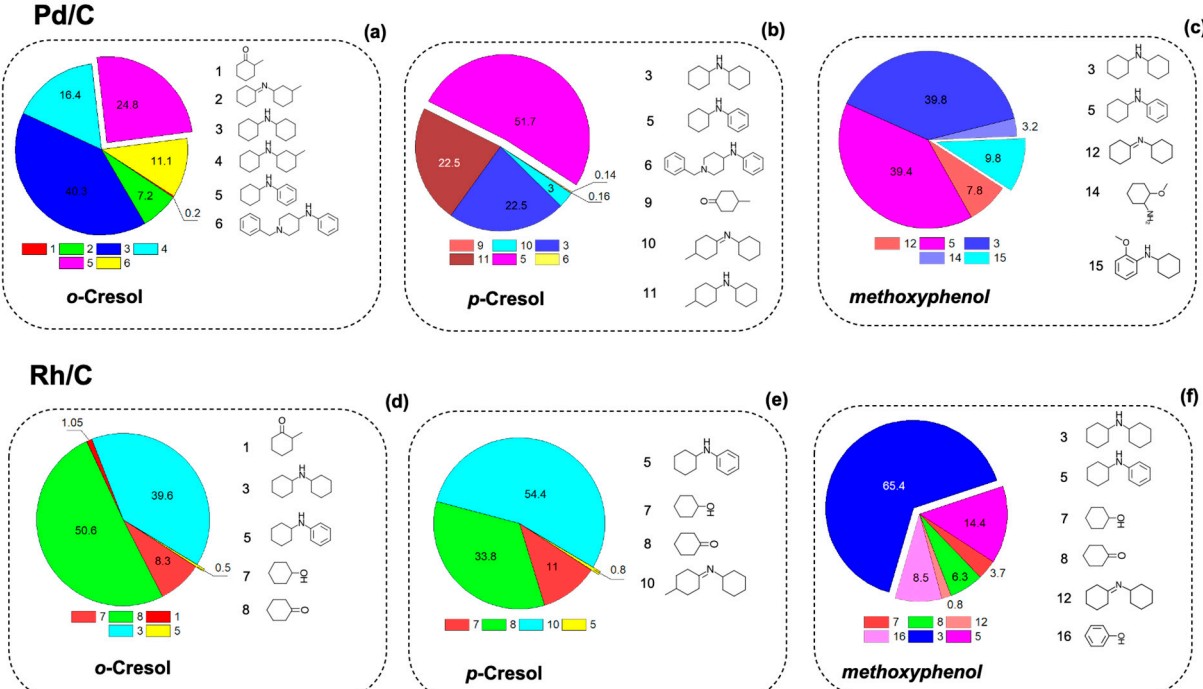

**Figure 6.** Influence of the type and position of substituents on the product distribution obtained from the phenolics reductive amination reaction. Values in the pies are referred to product concentrations (mol/L). (**a**–**c**) Pd/C (**d**–**f**) Rh/C. The reactions were carried out using $H_2$ gas (1.5 bar), $C^0_{Phenolic}$ = 0.2 mol/L, CyA (1.5 Eq to PhOH) and 120 °C with $t_R$ = 6 h. Details on product identification can be found in ESI_1.

The cresol isomers, o-cresol and p-cresol, exhibited similar conversion rates of 23% and 26%, respectively. These isomers also exhibited comparable selectivity towards their corresponding methylcyclohexylamines with 16.4% and 22.5%, respectively. In contrast, the -$CH_3O$ group in the methoxyphenol proved to be more active than the -$CH_3$ in cresols, as

indicated by its higher conversion rate ($X_{MePhOH} > X_{p\text{-}cresol} > X_{o\text{-}cresol}$) and intrinsic activity ($TOF_{MePhOH}$ (3.37 h$^{-1}$) > $TOF_{p\text{-}cresol}$ (1.41 h$^{-1}$) > $TOF_{o\text{-}cresol}$ (1.25 h$^{-1}$)) depicted in Figure 7a,b.

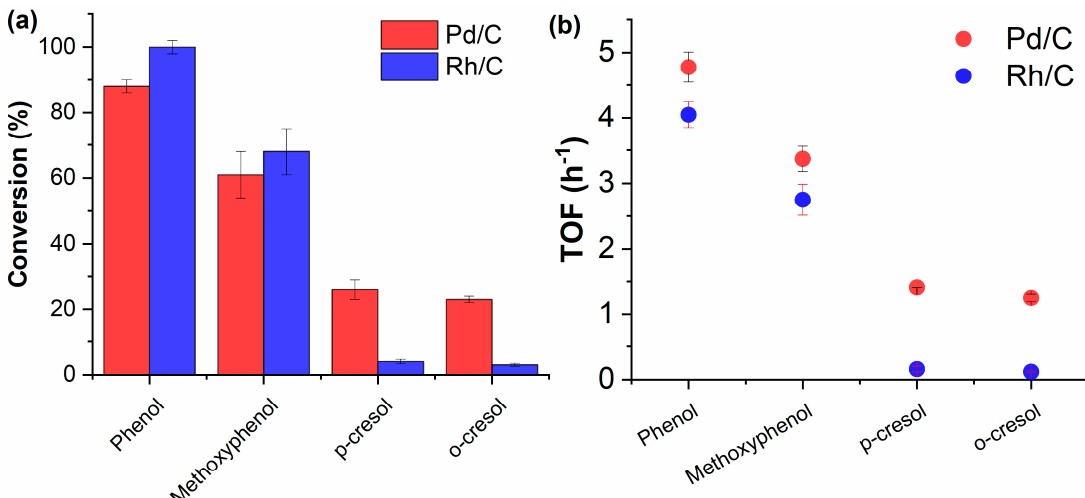

**Figure 7.** (**a**) TOF (h$^{-1}$) and (**b**) Conversion for substituted phenols on Rh/C and Pd/C. The TOFs are calculated from the average reaction rates of phenolics conversion curves. The reactions were carried out using H$_2$ gas (1.5 bar), C$^0_{Phenolic}$ = 0.2 mol/L, CyA (1.5 Eq to PhOH) and 120 °C with t$_R$ = 6 h.

The Rh/C catalyst did not produce any evidence of forming substituted secondary amines from the phenolics. Instead, Rh/C proved to be effective in hydrogenating these compounds into their corresponding ketones such as cyclohexanone, 2-methylcyclohexanone, and 4-methylcyclohexanone, and cyclohexanol, with selectivity ranging from 3.7% to 11% for the alcohol. The secondary amines that were formed over Rh/C were dicyclohexylamine and N-cyclohexylaniline, indicating the auto-condensation of cyclohexylamine, which was previously proposed.

The results obtained for cresols over Pd/C and Rh/C catalysts indicate that the conversion of cresols into their respective alkylated amines or ketones is independent of the position of the -CH$_3$ group. The TOF values for these compounds (Figure 7b) suggest that they undergo conversion via a similar reaction route. This observation is consistent with previous work by Ono et al. [36] who demonstrated that ring hydrogenation is the rate-determining step when cresols undergo amination over group VIII metals [37].

For methoxyphenol, the conversion to secondary amines over Pd was higher than that for alkylated cresols, indicating that the -CH$_3$OH substituted phenol is more easily hydrogenated than those containing -CH$_3$. The differences between these groups could also be attributed to the formation of side products that may hinder the selectivity of desirable products by increasing the catalyst's surface coverage. However, this assumption requires confirmation by measuring the initial reaction rates in the presence of side products.

The above results demonstrate that the position and nature of the substituent can hinder the hydrogenation of phenolic compounds and reduce their conversion into desirable products. Therefore, the reaction could be performed in two stages by separating phenol hydrogenation from amination. Thereafter, the carbonyl compound and the amine could form the intermediary imine, which is further hydrogenated into the final product.

## 3. Materials and Methods

### 3.1. Catalysts and Reagents

Commercial Pd/C and Rh/C catalysts at 10 wt.% (CAS-87104) and 5 wt.% (CAS-7440-16-6) in their oxide form were acquired from Merck (Santiago, Chile) and Alfa-Aeser (Chile), respectively. Before use (in characterization and reaction), catalysts were dried, sieved, and treated ex-situ at 400 °C (heating rate of 1.5 °C/min) for 2 h under a H$_2$ flow (40 mL/min) in a U-shaped fixed-bed reactor.

All the reagents were acquired at their analytical degree and used without further purification: Phenol (CAS-108-95-2, >99%), *o*-cresol (CAS-95-48-7, >99%), *p*-cresol (CAS-106-44-5, >98%), methoxyphenol (CAS-90-05-1, >98%), cyclohexylamine (CAS-108-91-8, >99%), $H_2$ (>99.999%, Iconsa, Santiago, Chile) and tert-amyl alcohol (CAS-75-85-4, >99%) as solvent were purchased from Merck Group (Chile). Details for secondary amines can be found in a previously published paper [38].

### 3.2. Catalyst Characterization

The catalysts were characterized by a set of complementary techniques to inspect their composition, morphology, structural properties, etc. This characterization included: Nitrogen adsorption-desorption isotherms measured at 77 K on a Micromeritics 3-Flex instrument (Micromeritics Inst. Co., Norcross, GA, USA). X-ray diffraction patterns in a Bruker D4 diffractometer (Bruker Co., Leipzig, Germany) with a $CuK_\alpha$ radiation source ($\lambda$ = 0.154059 nm). Moreover, metal particle morphology and particle size distribution were determined from transmission electron microscopy (JEOL JEM-1200 EXII equipment). In addition, a JEOL JEM-2200FS microscope (JEOL, Mitaka, Japan) with double aberration corrected was used to obtain high-resolution transmission electron microscopy (HRTEM) images and electron diffraction patterns. A detailed description of the procedures and analytical techniques used for catalysts characterization can be gathered from previous published papers [22–24]. X-ray photoelectron spectroscopy (XPS) was used to determine the chemical status of Pd and Rh in the catalysts before and after reduction. Spectra were recorded in vacuum (10-9 mbar) and 25 °C with a SPECS 150 MCD 9 PHOIBOS analyzer (SPECS, GmbH, Humbolthain, Germany) and Alk$\alpha$ monochromatic X-ray energy of 1486.60 eV. The binding energy was corrected using C as a reference component located at 284.6 eV. Surface alteration of graphite, graphite monofluoride and teflon by interaction with Ar+ and Xe+ beams. Applications of Surface Science, 1(4), 503–514.). Data analysis was performed with CasaXPS software (Casa Software Ltd.) using the Handbook of X-ray Photoelectron Spectroscopy [39], where the surface spectra of the catalysts were taken in situ before and after reduction to compare the changes in the chemical state of the Pd and Rh species, respectively.

### 3.3. Catalytic Conversion of Phenols

Liquid-phase reactions between the phenolics and cyclohexylamine, were carried out in 4 mL reinforced glass autoclave reactors inserted in a Reacti-ThermTM system (Thermofisher, Waltham, MA, USA) equipped with an external temperature probe, magnetic stirring (900 rpm) and pressurization line according to a procedure reported elsewhere [7,24]. Refs. [14,16,19,40] Prior to the activity tests, the reactor underwent three purges with helium gas (Air Liquide, Santiago, Chile, 99.99%). The reactor was then pressurized with hydrogen gas to the desired level. Mass flow controllers (Aalborg GFC17, Orangeburg, NY, USA) were used to regulate the gas flows. The resulting reaction products were filtered using 0.45 μm Teflon filters and subsequently analyzed ex situ via gas chromatography.

#### 3.3.1. Product Identification

The products were analyzed by means of a gas chromatograph (Clarus 690, PerkinElmer, Waltham, MA, USA) that was equipped with a mass spectrometer (QS8, Perkin-Elmer). The separation of chemical species was achieved using an Elite 1701 column (30 m × 0.25 mm × 0.25 μm) with He as the carrier gas at a flowrate of 15 mL min$^{-1}$. The oven temperature was ramped from 45 to 280 °C at a rate of 2.5 °C min$^{-1}$. The compounds were identified by comparing their ionization patterns (*m/z* range: 30–600 Da) with the standard spectra database of the NIST library [7,24,38]. The results of the GC/MS are presented in the Electronic Supplementary Information (ESI_1) and in a Mendeley data repository: https://data.mendeley.com/datasets/z5bypjm62y/1 (accessed on 24 March 2023).

Conversion of reactants (amine and phenol) and selectivity of products (based on GC-FID measurements) were defined as: (initial moles of reactant−final moles of reactant)/initial

moles of reactant × 100), and (moles of product i)/moles of total products × 100, respectively. Turnover frequency (TOF) was defined as: (moles of product i per time unit)/(moles of surface metal in the catalyst).

### 3.3.2. Products Quantification

The products were quantified using a gas chromatograph (8610, SRI Instruments, Torrance, CA, USA) equipped with an on-column injection port and a flame ionization detector. The separation of analytes was achieved with an MTX-5 column (30 m × 0.25 mm × 0.1 μm) and the retention times were assigned by injecting solvent-pure samples prior to the product analysis. The oven heating program was as follows (Table 3):

**Table 3.** Oven temperature program used for GC analysis.

| Initial Temperature | Hold | Ramp | Final Temperature |
|:---:|:---:|:---:|:---:|
| 45 °C | 1 min | 1.5 °C min$^{-1}$ | 50 °C |
| 50 °C | 1.5 min | 2 °C min$^{-1}$ | 100 °C |
| 100 °C | 5 min | 10 °C min$^{-1}$ | 250 °C |

The quantification was performed using nonane as an internal standard.

The original files with the quantification data can be found in https://data.mendeley.com/datasets/z5bypjm62y/1 (accessed on 24 March 2023).

The reported values are the means of three replicates, and their corresponding standard deviations (Stdev) are shown in the figures. The Stdev was calculated using an Excel function as follows: $\sqrt{\dfrac{\Sigma\left(x-\bar{x}\right)^2}{(n-1)}}$.

### 4. Conclusions

In conclusion, this study demonstrated that Pd/C and Rh/C catalysts have suitable textural and structural properties to achieve significant yields in the reductive amination of phenol. The analysis of the reaction conditions and kinetic approach identified a similar reaction pathway for both catalysts involving reduction, condensation, and disproportionation. Self-condensation-dehydrogenation of cyclohexylamine was observed, leading to the formation of dicyclohexylamine and cyclohexylaniline. Batch tests confirmed the capability of Pd and Rh to hydrogenate the intermediary imine, leading to the predominant formation of dicyclohexylamine. However, Pd/C produced cyclohexylaniline at low $H_2$ availability and high temperatures due to over-dehydrogenation of the imine. The power law kinetic model indicated negligible coverage of phenol on the surface, and a possible direct dissociation of $H_2$ participated in the reaction mechanism. Furthermore, the study revealed a higher coverage of cyclohexylamine, leading to its participation in a parallel reaction pathway through self-condensation. Finally, experiments with substituted phenols demonstrated that the nature of the substituent influences the hydrogenation of the phenol and its subsequent amination. These findings provide valuable insights into the reductive amination of phenol, highlighting the importance of catalyst selection and reaction conditions for achieving high yields.

**Supplementary Materials:** The following supporting information can be downloaded at: https://www.mdpi.com/article/10.3390/catal13040654/s1, Figure S1: Transmission electron microscopy images and mean particle size (dp) of (a) Pd/C, (b) Rh/C after reduction; Figure S2: Stability catalytic assays (a) Pd/C and (b) Rh/C. T = 120 °C, $P_{H2}$ = 1.5 bar, $C^0_{PhOH}$ = 0.2 mol/L, $C^0_{Amine}$ = 1.4 Eqv., t = 6 h, $V_R$ = 4 mL; Figure S3: Arrhenius plots for reductive phenol amination with cyclohexylamine (a) reaction performed on Pd/C, (b) reaction performed on Rh/C; Figure S4: Fitting plots of power law kinetic model data fit to determine reaction orders of (a), (d) cyclohexylamine (CyA); (b), (e), Hydrogen ($H_2$) and (c), (f) phenol (PhOH) of Pd/C and Rh/C, respectively; Table S1: Binding energy (BE, eV), kinetic energy (KE, eV) and modified Auger parameter ($\alpha'$ = BE + KE, eV) values of Pd

3d$_{5/2}$ and Rh 3d$_{5/2}$ on catalysts before reduction the H$_2$ and after reduction conditions (400 °C and H$_2$); Table S2: Amination the phenol of different amines catalyst Pd/C and Rh/C in presence of H$_2$ (1.5 bar). Reactions were carried in 10 mL glass autoclaves, 0.2 mol/L of phenol, using tert- amyl alcohol as a solvent.

**Author Contributions:** Conceptualization, L.E.A.-P. and R.J.; methodology, M.O., M.E.D. and R.M.; formal analysis, M.O., M.P. and R.M.; investigation, M.O., M.P. and R.M.; resources, L.E.A.-P. and R.J.; data curation, R.M. and M.O.; writing—original draft preparation, R.M. and M.O.; writing—review and editing, M.O., L.E.A.-P., M.E.D. and R.J.; supervision, L.E.A.-P. and R.J.; project administration, L.E.A.-P. and M.O.; funding acquisition, L.E.A.-P., M.E.D. and R.J. All authors have read and agreed to the published version of the manuscript.

**Funding:** This research was funded by Agencia Nacional de Investigaciones (Chile), grant number FONDECYT 1190063 and the University of Bio-Bio through the internal project 2260311 AD/EQ.

**Data Availability Statement:** All the data used in the paper is available in a Mendeley Data repository (https://data.mendeley.com/datasets/z5bypjm62y/1 (accessed on 24 March 2023) and in a Data in Brief paper [24].

**Conflicts of Interest:** The authors declare no conflict of interest.

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
