# Peer review of "Secondary Amines from Catalytic Amination of Bio-Derived Phenolics over Pd/C and Rh/C: Effect of Operation Parameters"

_catalysts, doi:10.3390/catal13040654_

Round 1
Reviewer 1 Report
The article with the title: “Secondary amines from catalytic amination of bio-derived phenolics over Pd/C and Rh/C: Effect of operation parameters.” describes a set of reactions with either Pd/C or Rh/C to form secondary amines. In the introduction the authors indicate, that renewable resources like lignocellulosic biomass can be applied to get monomeric structures after depolimerization. Different catalysts which are used for hydrogenation are described. The authors presented the characterization of the two used catalysts and described the outcome of the experiments in the results section. The results are discussed in the same part of the article. The materials and methods section is short, and most of the procedures are not fully described but references are given, which is a bit confusing.
General comments:
-The Materials and Methods section should be section 2 after the introduction.
-The Discussion section is missing. Maybe think about renaming the Results section to Results and Discussion or provide a separate Discussion section after the Results.
- Please check if all Company names were (e.g. Mass spectrometer) are given in the article.
-The exact procedures for the amination process are missing.
-How was the quantification of the products done and how did the authors determine total conversion of the educts? Please describe in detail.
Specific comments:
-line 287: 0,15 0 what does it mean?
-line 315 The meaning of 5% and 10% Me is not described.
-line 460 2methyl should be 2-methyl
Author Response
Concepción, March 2023
Dear Reviewer
On behalf of the authors, I would like to thank you for your valuable comments and criticisms on our manuscript entitled “Secondary amines from catalytic amination of bio-derived phenolics over Pd/C and Rh/C: Effect of operation parameters”. Here we provide an itemized list of the changes and corrections made in response to your evaluation.
With kind regards,
Corresponding Author
Prof. Dr. L.E., Arteaga-Pérez,
Laboratory of Thermal and Catalytic Processes (LPTC-UBB).
Chile
E-mail: larteaga@ubiobio.cl
REVIEWER-1
Comments and Suggestions:
The article with the title: “Secondary amines from catalytic amination of bio-derived phenolics over Pd/C and Rh/C: Effect of operation parameters.” describes a set of reactions with either Pd/C or Rh/C to form secondary amines. In the introduction the authors indicate, that renewable resources like lignocellulosic biomass can be applied to get monomeric structures after depolimerization. Different catalysts which are used for hydrogenation are described. The authors presented the characterization of the two used catalysts and described the outcome of the experiments in the results section. The results are discussed in the same part of the article. The materials and methods section is short, and most of the procedures are not fully described but references are given, which is a bit confusing.
Thank you for your comment. After taking into account your observation, as well as those of other reviewers, we have made some modifications to the materials and methods section. Specifically, we have included two new sections that describe the procedures for conducting reactions and chromatographic analyses (using both GC/MS and GC/FID techniques). Additionally, we have added a new electronic supplementary material (ESI_2) which provides GC/MS data for compound identification and quantification. We have also maintained references to a Mendeley data repository and a Data in Brief paper as complementary sources.
General comments:
-The Materials and Methods section should be section 2 after the introduction.
We agree with the referee that the current ordering may be a bit confusing. However, we would like to clarify that this particular ordering was necessary in order to comply with the format requirements of the journal. Specifically, the Journal's template was used to prepare the manuscript, and as such, we are unable to rearrange the manuscript.
-The Discussion section is missing. Maybe think about renaming the Results section to Results and Discussion or provide a separate Discussion section after the Results.
We have renamed the Results section as Results and Discussion.
- Please check if all Company names were (e.g. Mass spectrometer) are given in the article.
All company names, country and models have been included for the analytical devices.
-The exact procedures for the amination process are missing.
As previously mentioned, we have included a section describing the procedures for reactions and chromatographic analysis (GC/MS and GC/FID).
-How was the quantification of the products done and how did the authors determine total conversion of the educts? Please describe in detail.
The products were quantified using a gas chromatograph (8610, SRI Instruments, USA) equipped with an on-column injection port and a flame ionization detector (FID). The separation of analytes was achieved with an MTX-5 column (30 m x 0.25 mm x 0.1 μm) and the retention times were assigned by injecting solvent-pure samples prior to the product analysis. The oven heating program was as follows:
|
Initial Temperature |
Hold |
Ramp |
Final Temperature |
|
45 ºC |
1 min |
1.5 ºCmin-1 |
50 ºC |
|
50 ºC |
1.5 min |
2 ºCmin-1 |
100 ºC |
|
100 ºC |
5 min |
10 ºCmin-1 |
250 ºC |
The quantification was performed using nonane (CAS-111-84-2, >99%) as an internal standard. The chemical standards used for calibration were:
phenol (CAS-108-95-2, >99%), o-cresol (CAS-95-48-7, >99%), p-cresol (CAS-106-44-5, >98%), methoxyphenol (CAS-90-05-1, >98%), cyclohexylamine (CAS-108-91-8, >99%), tert-amyl alcohol (CAS-75-85-4, >99%), N-cyclohexaneimine (CAS-1132-38-3, >95%), cyclohexanone (CAS-108-94-1, >99%), Dicyclohexylamine (CAS-101-83-7, >99%), Cyclohexylaniline (CAS-1821-36-9, >98%) and Diphenylamine (CAS-122-39-4, >99%).
The original GC files with the quantification data can be found in Manrique Suárez, Raydel; Ortega Díaz, Maray; Arteaga Pérez , Luis E. ; Garrido, Benjamin (2022), “Dataset on the characterization, and application of Pd and Rh-based catalysts in the reductive amination of biomass-derived phenolics”, Mendeley Data, V1, doi: 10.17632/z5bypjm62y.1.
The conversion of reactants (amine and phenolics) were calculated based on GC-FID measurements as: (initial moles of reactant−final moles of reactant)/initial moles of reactant×100).
This information is now available in the revised manuscript and ESI.
Specific comments:
-line 287: 0,15 0 what does it mean?
This was a typing error, now it was corrected as 0.15.
-line 315 The meaning of 5% and 10% Me is not described.
These were the metal content in both catalysts, now the values were removed from this point to avoid misunderstandings.
-line 460 2methyl should be 2-methyl
This was corrected.

Reviewer 2 Report
The following .pdf file contains comments and suggestions for the authors.

Author Response
Concepción, March 2023
Dear Reviewer
On behalf of the authors, I would like to thank you for your valuable comments and criticisms on our manuscript entitled “Secondary amines from catalytic amination of bio-derived phenolics over Pd/C and Rh/C: Effect of operation parameters”. Here we provide an itemized list of the changes and corrections made in response to your evaluation.
With kind regards,
Corresponding Author
Prof. Dr. L.E., Arteaga-Pérez,
Laboratory of Thermal and Catalytic Processes (LPTC-UBB).
Chile
E-mail: larteaga@ubiobio.cl
REVIEWER-2
Brief summary:
The authors describe using precious metals supported on carbon to perform catalytic amination of biomass-derived phenols to generate secondary amines. This work is relevant to the conversion of biomass to value-added products as humans try to ease their dependence on impermanent fossil fuels. The authors report high-resolution transmission electron microscopy and X-ray photoelectron spectroscopy to characterize the surface of the Pd/C and Rh/C catalysts. The authors perform catalytic experiments that vary the cyclohexylamine, dihydrogen pressure, and temperature to glean information about the effects operational parameters have on the conversion and selectivity of secondary amine products. From the operational parameters, the authors draw conclusions from the data using a “power law” kinetic modeling approach and propose a mechanism. Additionally, steric effects are investigated to determine the impact on conversion and turn over frequency.
After the following major points are addressed, this article should be reconsidered for publication.
General comments:
1) The current manuscript lacks experimental evidence for the conversion of starting materials to the various products shown. In the materials and methods section, there is a subsection about the catalytic conversion of phenols, but no gas chromatography or mass spectrometry data are provided here to demonstrate that these complexes were truly observed. Please provide this data in the supporting information.
Thank you for your comment. We appreciate your observations, which were also noted by other reviewers. Based on this feedback, we have revised the materials and methods section to include a new subsection outlining the procedures for amination reactions and chromatographic analyses (GC/MS and GC/FID). Additionally, we have included new electronic supplementary material that provides GC/MS data for compound identification. Furthermore, the original GC-FID files, along with the quantification data, have been made available in a Mendeley Data repository.: Manrique Suárez, Raydel; Ortega Díaz, Maray; Arteaga Pérez , Luis E. ; Garrido, Benjamin (2022), “Dataset on the characterization, and application of Pd and Rh-based catalysts in the reductive amination of biomass-derived phenolics”, Mendeley Data, V1, doi: 10.17632/z5bypjm62y.1.
2) Error analysis and error bars are not present in the figures or tables that make claims about conversion and selectivity. This is critical to evaluate the authors’ claims about the impact of the operational conditions on the catalytic conversion and yield of the products.
We appreciate your feedback regarding the error bars for the replicates in our experiments. We agree that these were mistakenly omitted from the original Figures 3, 5, and 7. We have now corrected this oversight by updating the figures to include the error bars for the two or three replicates performed in each experiment. Thank you for bringing this to our attention.
3) The reaction mechanism referred to in the manuscript on line 426 and presented in Figure S1 of the supporting information is proposed with almost no supporting evidence or discussion in the manuscript. The Figure S1 caption should be updated since a) and b) panels are shown but are not referred to. Additionally, if the authors maintain the proposed mechanism, a detailed discussion should be included to support the proposal.
We appreciate your feedback on the proposed mechanism in our manuscript. We acknowledge that, at this stage, the mechanism is speculative and requires further specific measurements to support the proposed series of elementary steps. In light of this, we have decided to eliminate Figure S1 and any references to the mechanism from the manuscript.
However, we would like to emphasize that this work is still ongoing, and we are currently conducting FTIR-DRIFT and FTIR-In situ measurements to support our proposed mechanism. We are also carrying out a mechanism-based kinetic analysis to further validate our findings. We hope that these additional experiments will provide a more thorough understanding of the mechanism in question.
4) The fitting of the data from the power law kinetic modeling should be presented in the supporting information. Additionally, the residual plots of the data fitting should be provided in the supporting information to support their claims about the reaction orders for the reactants. This will benefit others who read the manuscript too.
5) The authors should provide an Arrhenius plot to support their claim that an Arrhenius analysis was performed.
Considering the recommendations in comments 4 and 5, we have included the Arrhenius plots and data fitting for reaction orders in the supporting information. Moreover, we have made a reference to these figures in the revised manuscript.
6) If an Arrhenius analysis was performed, presumably an Eyring analysis can be performed here too. Have the authors thought about performing Eyring analysis as well to learn about the enthalpy and entropy of activation for these transformations?
The authors have considered the Eyring analysis; however, the Power Law models obtained in this study represent a comprehensive portrayal of a complex multi-reaction mechanism. Consequently, the estimated activation energies are apparent values, and the ΔH and ΔS obtained from Eyring's analysis are not indicative of the physical phenomena occurring in each individual reaction step. Therefore, a mechanism-based approach (currently underway) will be used to estimate these parameters accurately.
7) The authors should clarify what H* means in the manuscript. Is this an adsorbed hydrogen atom, proton, or hydride?
This is an absorbed hydrogen, now it is mentioned in the revised manuscript.
Specific comments:
1) Page 1, line 41, please add the following citation: Salvatore, R.N.; Yoon, C.H.; Jung, K.W. Synthesis of secondary amines. Tetrahedron 2001, 57, 7785-7811, doi: https://doi.org/10.1016/S0040-4020(01)00722-0.
Done
2) It is unclear how citation 2 benefits the opening sentence in the introduction; secondary amines are hardly mentioned in this book, which is generally about laboratory accidents. If the authors are suggesting that one of the reagents being used is particularly hazardous this should be addressed directly. On the other hand, if they are calling attention to a dangerous process that they are making safer, they should say that. Otherwise, the citation should be replaced with something more relevant.
This was an error. Now the references used here are:
[1] Royer J. Chiral Amine Synthesis. Methods, Developments and Applications. Edited by Thomas C. Nugent. Angewandte Chemie International Edition 2010;49:7841–7841. https://doi.org/10.1002/anie.201005721.
[2] Salvatore, R. N., Yoona, C. H., Junga, K. W, et al. Synthesis of secondary amines. Tetrahedron 2001;57,7785-7811. http://doi.org/10.1016/S0040-4020(01)00722-0.
3) Reference 22 is incomplete.
Corrected
4) The sentence ending on line 47 needs a citation.
We have included the following cites:
- Zakzeski J, Bruijnincx PCA, Jongerius AL, Weckhuysen BM. The catalytic valorization of lignin for the production of renewable chemicals. Chem Rev 2010;110:3552–99. https://doi.org/10.1021/cr900354u.
- Hiltunen J, Kuutti L, Rovio S, Puhakka E, Virtanen T, Ohra-Aho T, et al. Using a low melting solvent mixture to extract value from wood biomass. Sci Rep 2016;6. https://doi.org/10.1038/srep32420.
- Rahimi A, Azarpira A, Kim H, Ralph J, Stahl SS. Chemoselective metal-free aerobic alcohol oxidation in lignin. J Am Chem Soc 2013;135:6415–8. https://doi.org/10.1021/ja401793n.
- Lancefield CS, Ojo OS, Tran F, Westwood NJ. Isolation of functionalized phenolic monomers through selective oxidation and CO bond cleavage of the β-O-4 linkages in Lignin. Angewandte Chemie - International Edition 2015;54:258–62. https://doi.org/10.1002/anie.201409408.
- Nguyen JD, Matsuura BS, Stephenson CRJ. A photochemical strategy for lignin degradation at room temperature. J Am Chem Soc 2014;136:1218–21. https://doi.org/10.1021/ja4113462.
- Deuss PJ, Scott M, Tran F, Westwood NJ, de Vries JG, Barta K. Aromatic Monomers by in Situ Conversion of Reactive Intermediates in the Acid-Catalyzed Depolymerization of Lignin. J Am Chem Soc 2015;137:7456–67. https://doi.org/10.1021/jacs.5b03693.
5) There are several minor grammatical errors and/or misspellings. The following is a list that may not be all-inclusive: lines 65, 74, 75, 93, 121, 130, 177, 295, 309,394, 415, 422, 439, 443, 460, 538.
These errors were corrected, and the English was thoroughly revised.
6) Lines 344-346 should be removed because the authors are speculating.
Done
7) Lines 388-389 have a different font/font size compared to lines 385-386.
Corrected.
8) On line 452, the authors refer to methoxy groups as “electron-withdrawing.” This should be updated to specify that a methoxy group is inductively withdrawing due to the oxygen atom. A methoxy group is an electron-donating group by resonance.
The authors thank the referee for this observation. Accordingly, this entire section was rewritten, and this error was corrected. Moreover, the authors have been less speculative in the new discussion and in the manuscript conclusions.
9) Line 537, …participation by H… is this H2, a proton, hydrogen atom, or hydride?
This is a hydrogen atom, now it was defined in the text.
10) The title of section 2.1 is “Characterization of Pd/C and Rh/C catalysts” and then lines 120-121 states that extensive characterization can be found in another paper from their group. The section should be renamed or addressed to reflect the types of characterization carried out in the current manuscript.
The section is renamed as: Surface, textural and morphologic properties of catalysts.
11) In Figure 4, and throughout the manuscript and SI, “equilibrium” arrows should be shown instead of the “resonance” arrows that are currently presented.
Corrected.

Reviewer 3 Report
Reductive amination of bio-derived phenolics to secondary amines is of great significance. The authors have reported that the Pd/C and Rh/C catalysts are suitable to achieve significant yields in the reductive amination of phenol. In addition, a systematic analysis on the operational parameters and their kinetic implications were included in the analysis. The study of reaction pathways and operational conditions is comprehensive, which will have a significant impact on the subsequent reduction amination of biomass. The incentive of the research is stated clearly and the experiments are well-planned. This manuscript can be accepted after these minor revisions are carried out:
1. The authors have claimed that the catalysts have similarities in acid properties. As we can see from the table 1, the total acidity between Pd/C and Rh/C catalysts is largely different. So, what is the effect of acidity on reductive amination.
2. The caption of Figure 1 indicated that the size distribution diagrams were in Supporting Information. But I do not find it in Supporting Information. I want to see particle size distribution about Pd/C and Rh/C catalysts.
3. It is clear that metal sites (such as Pd, Rh) play important roles in hydrogenation/dehydrogenation steps. However, it is easy to inactivate during amination process. The stability experiments of catalysts should be supplemented.
Author Response
Concepción, March 2023
Dear Reviewer
On behalf of the authors, I would like to thank you for your valuable comments and criticisms on our manuscript entitled “Secondary amines from catalytic amination of bio-derived phenolics over Pd/C and Rh/C: Effect of operation parameters”. Here we provide an itemized list of the changes and corrections made in response to your evaluation.
With kind regards,
Corresponding Author
Prof. Dr. L.E., Arteaga-Pérez,
Laboratory of Thermal and Catalytic Processes (LPTC-UBB).
Chile
E-mail: larteaga@ubiobio.cl
REVIEWER-3
Brief summary:
Reductive amination of bio-derived phenolics to secondary amines is of great significance. The authors have reported that the Pd/C and Rh/C catalysts are suitable to achieve significant yields in the reductive amination of phenol. In addition, a systematic analysis on the operational parameters and their kinetic implications were included in the analysis. The study of reaction pathways and operational conditions is comprehensive, which will have a significant impact on the subsequent reduction amination of biomass. The incentive of the research is stated clearly and the experiments are well-planned. This manuscript can be accepted after these minor revisions are carried out:
- The authors have claimed that the catalysts have similarities in acid properties. As we can see from the table 1, the total acidity between Pd/C and Rh/C catalysts is largely different. So, what is the effect of acidity on reductive amination.
In fact, there was an error in the total acidity data reported in Table 1. This error was corrected, and the actual values are (8.9±0.4)10-3 µmol NH3/m2g (Rh/C) and (14.1±2.3)10-3 µmol NH3/m2g (Pd/C), which are in the same order of magnitude.
- The caption of Figure 1 indicated that the size distribution diagrams were in Supporting Information. But I do not find it in Supporting Information. I want to see particle size distribution about Pd/C and Rh/C catalysts.
Thanks for letting us know about this omission, now the particle size distribution diagrams and TEM images are reported in the supplementary information (Fig. S1).
- It is clear that metal sites (such as Pd, Rh) play important roles in hydrogenation/dehydrogenation steps. However, it is easy to inactivate during amination process. The stability experiments of catalysts should be supplemented.
The catalytic stability of fresh Pd/C and Rh/C was evaluated as part of ongoing research (See Figure below). However, there are some characterization analyses still pending, thus we decided not to include these results in the present manuscript.
Considering your suggestion, we have included a Figure in the supplementary material and a short discussion in the revised manuscript.
The catalytic performance of Pd/C and Rh/C was evaluated over four and six cycles of reductive amination, respectively. Pd/C remained stable for the first three cycles, after which its activity sharply decreased, while Rh/C showed a similar pattern, with activity decreasing between the 5th and 6th cycles. The observed reduction in activity could be due to various factors such as metal leaching, carbon deposition, or sample loss between cycles. Additionally, since pre-reduction was not performed between cycles, any oxidation of the catalyst surface during manipulation (washing-drying) may have led to deactivation of Pd0 and Rh0 sites.
To investigate the possibility of metal leaching, additional ICP-OES elemental analysis was conducted. The results revealed that there was no significant variation in metal content for both Pd/C and Rh/C.
|
Catalysts |
After 1er use |
After 3er use |
After regeneration |
|
Pd/C |
10.76 |
9.93 |
9.37 |
|
Rh/C |
5.98 |
5.01 |
4.65 |
In addition, we performed scanning transmission electron microscopy (STEM-EDS) for Pd/C and Rh/C (under development) and confirmed that after reaction, the Pd atoms were homogeneously distributed on the catalyst surface. Currently, the XPS analysis of used Pd/C and Rh/C are being carried out to elucidate if their elemental chemical state and composition remain unchanged.

Reviewer 4 Report
In this manuscript, the authors present a detailed investigation into the catalytic amination of phenolics derived from lignocellulosic biomass to secondary amines using Pd/C and Rh/C. The operational parameters were systematically explored. This is the first study addressing the operational parameters on the catalytic amination of phenols. The investigation involved characterization of the used Pd/C and Rh/C catalysts through HRTEM imaging and XPS spectra, control experiments of the reactant concentrations and temperature, as well as steric and electronic effects of the phenolics. The investigation was performed at rigorous standards. The results will be valuable to chemists and engineers in optimizing amination reactions from phenolics. The only potential concern is in the comparison of the methyl and methoxy substituted phenols. The trend in conversion and TOF between phenol, methoxyphenol, and cresol is not intuitive. The authors indicate that the methoxy group is electron-withdrawing, while methoxy groups are usually considered as electron-donating. The authors further attempt to use steric hindrance of the methyl groups in cresol to explain the reduced conversions. The Me group in p-cresol is very distant from the OH group and should thus have no steric influence and performs equally as poor as the more hindered o-cresol. The role of hydrogen bonding is also not at all clear between these groups as the authors try to explain. Perhaps the it would be prudent to make less definitive statements regarding the role of sterics, electronics, and H-bonding within the substituted phenolics as the trends are not clear and counterintuitive in some cases.
The manuscript could use another round of proofreading. A few comments are provided below.
-line 46. change "synthesis methods" to "synthetic methods"
-line 93: change "hysteric" to "steric"
-line 120: "extense" to "extensive"
-line 199: lowercase "Carbon"
-line 204: cyclohexylamine spelling
-line 237: effectiveness
-line 378: differentiation
-line 414: formed
-line 452: methoxy groups are normally strongly electron donating, not withdrawing as written.
Author Response
Concepción, March 2023
Dear Reviewer
On behalf of the authors, I would like to thank you for your valuable comments and criticisms on our manuscript entitled “Secondary amines from catalytic amination of bio-derived phenolics over Pd/C and Rh/C: Effect of operation parameters”. Here we provide an itemized list of the changes and corrections made in response to your evaluation.
With kind regards,
Corresponding Author
Prof. Dr. L.E., Arteaga-Pérez,
Laboratory of Thermal and Catalytic Processes (LPTC-UBB).
Chile
E-mail: larteaga@ubiobio.cl
REVIEWER-4
Brief summary:
In this manuscript, the authors present a detailed investigation into the catalytic amination of phenolics derived from lignocellulosic biomass to secondary amines using Pd/C and Rh/C. The operational parameters were systematically explored. This is the first study addressing the operational parameters on the catalytic amination of phenols. The investigation involved characterization of the used Pd/C and Rh/C catalysts through HRTEM imaging and XPS spectra, control experiments of the reactant concentrations and temperature, as well as steric and electronic effects of the phenolics. The investigation was performed at rigorous standards. The results will be valuable to chemists and engineers in optimizing amination reactions from phenolics. The only potential concern is in the comparison of the methyl and methoxy substituted phenols. The trend in conversion and TOF between phenol, methoxyphenol, and cresol is not intuitive. The authors indicate that the methoxy group is electron-withdrawing, while methoxy groups are usually considered as electron-donating. The authors further attempt to use steric hindrance of the methyl groups in cresol to explain the reduced conversions. The Me group in p-cresol is very distant from the OH group and should thus have no steric influence and performs equally as poor as the more hindered o-cresol. The role of hydrogen bonding is also not at all clear between these groups as the authors try to explain. Perhaps it would be prudent to make less definitive statements regarding the role of sterics, electronics, and H-bonding within the substituted phenolics as the trends are not clear and counterintuitive in some cases.
The authors would like to express their gratitude to the referee for their insightful observation. As a result, we have rewritten this section to avoid speculation. We have also introduced new theories that are more cautious and based on the revised literature, as well as our own experimental observations.
The manuscript could use another round of proofreading. A few comments are provided below.
All the errors were corrected and an additional proof reading of the manuscript was performed.
-line 46. change "synthesis methods" to "synthetic methods"
Corrected
-line 93: change "hysteric" to "steric"
Corrected
-line 120: "extense" to "extensive"
Corrected
-line 199: lowercase "Carbon"
Corrected
-line 204: cyclohexylamine spelling
Corrected
-line 237: effectiveness
Corrected
-line 378: differentiation
Corrected
-line 414: formed
Corrected
-line 452: methoxy groups are normally strongly electron donating, not withdrawing as written.
Corrected. See the discussion in the revised manuscript.

Round 2
Author Response
Concepción, March 2023
Dear Reviewer
On behalf of the authors, I would like to thank you for your valuable comments and criticisms on our manuscript entitled “Secondary amines from catalytic amination of bio-derived phenolics over Pd/C and Rh/C: Effect of operation parameters”. Here we provide an itemized list of the changes and corrections made in response to your evaluation.
With kind regards,
Corresponding Author
Prof. Dr. L.E., Arteaga-Pérez,
Laboratory of Thermal and Catalytic Processes (LPTC-UBB).
Chile
E-mail: larteaga@ubiobio.cl
REVIEWER-2
Brief summary:
The authors describe using precious metals supported on carbon to perform catalytic amination of biomass-derived phenols to generate secondary amines. This work is relevant to the conversion of biomass to value-added products as humans try to ease their dependence on impermanent fossil fuels. The authors report high-resolution transmission electron microscopy and X-ray photoelectron spectroscopy to characterize the surface of the Pd/C and Rh/C catalysts. The authors perform catalytic experiments that vary the cyclohexylamine, dihydrogen pressure, and temperature to glean information about the effects operational parameters have on the conversion and selectivity of secondary amine products. From the operational parameters, the authors draw conclusions from the data using a power law kinetic modeling approach. Additionally, steric effects are investigated to determine the impact on conversion and turnover frequency.
The authors addressed my previous comments well, and after the following minor points are addressed, this article should be accepted for publication.
Specific comments:
1) Line 32, please change “confirmed” to “supported.”
Done
2) line 27, please change “easiness” to “propensity.”
Done
3) line 42, please change “specie” to “species.”
Done
4) Figures 3, 5, and 7 show error bars, but there is no explanation about where the error bars come from. The authors should include how they performed their error analysis.
The error bars indicate standard deviations for three replicates. This value is calculated from the excel function DESVEST, the details are provided in the revised version of the Materials and Methods section.
5) In Figure 5, panels C and F, please choose a different color for the error bars, they are currently difficult to see.
Colors in Fig. 5 were changed and now the error bars are visible.
6) In Figure 7, panel B, error bars should be included so that the reader may see if there is a difference between the turnover frequency for the conversion of products on Pd/C or Rh/C in the presence of the different solvents.
Done.
7) The authors should clarify what Figure S3 refers to. The data represents an Arrhenius plot, but for what process?
Done. Now the captions of the Fig. S3 reads as:
“Figure S3. Arrhenius plots for reductive phenol amination with cyclohexylamine (a) reaction performed on Pd/C, (b) reaction performed on Rh/C.”
8) In lines 38-43 the authors state…“reaction orders of 0.5 and 0.7 for H2 on Pd/C and Rh/C respectively” However, figure S4 appears to show reaction orders of 0.4 and 0.7 for H2 . Similarly, in the manuscript, phenol reaction order is listed between 1 and 1.9, but the data shows slopes of 1 and 1.8. Please make sure these are consistent.
These errors were removed from the revised manuscript, and now the values in the figures and the text are consistent.
9) The authors should provide error bars for the reaction orders proposed for cyclohexylamine, dihydrogen, and phenol based on their kinetic analysis shown in Figure S4.
Done.